# VIDEO PARALLEL SCALING: AGGREGATING DIVERSE FRAME SUBSETS FOR VIDEOLLMS

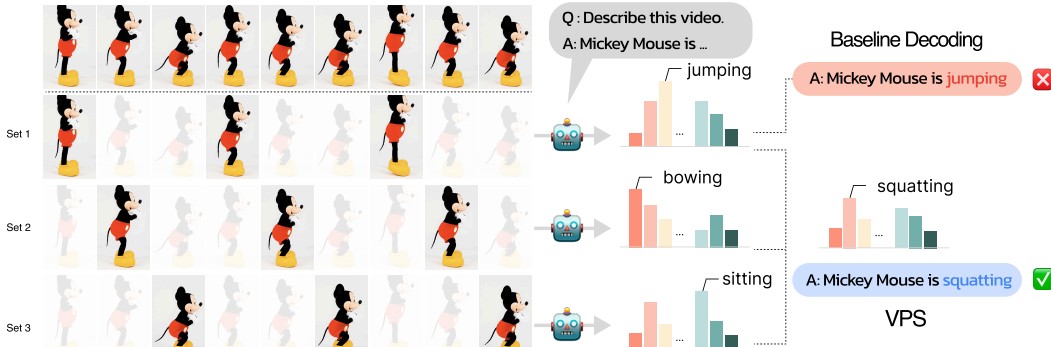

Figure 1: **Conceptual Illustration of VPS**. VideoLLMs take as input subsampled frames from the original video[1], limiting their understanding capabilities. Increasing the sampled frames within context leads to computation/memory issues or decrease in performance. In contrast, VPS keeps the number of subsampled frames, and scales the number of streams *in parallel*, with each stream attending to different frames. By aggregating the probability values from each stream, VPS results in 1) fine-grained motion perception, 2) consistent gains with more streams, without increasing the context length.

## ABSTRACT

Video Large Language Models (VideoLLMs) face a critical bottleneck: increasing the number of input frames to capture fine-grained temporal detail leads to prohibitive computational costs and performance degradation from long context lengths. We introduce Video Parallel Scaling (VPS), an inference-time method that expands a model's perceptual bandwidth without increasing its context window. VPS operates by running multiple parallel inference streams, each processing a unique, disjoint subset of the video's frames. By aggregating the output probabilities from these complementary streams, VPS integrates a richer set of visual information than is possible with a single pass. We theoretically show that this approach effectively contracts the Chinchilla scaling law by leveraging uncorrelated visual evidence, thereby improving performance without additional training. Extensive experiments across various model architectures and scales (2B-32B) on benchmarks such as Video-MME and EventHallusion demonstrate that VPS consistently and significantly improves performance. It scales more favorably than other parallel alternatives (e.g. Self-consistency) and is complementary to other decoding strategies, offering a memory-efficient and robust framework for enhancing the temporal reasoning capabilities of VideoLLMs.

## 1 INTRODUCTION

Video Large Language Models (VideoLLMs) inherit the cross-modal reasoning abilities of image-based VLMs, extending the models with multi-frame inputs. This access to temporal information enables them to (i) reason about causality and motion by tracing how an object's state evolves,

---

[1]Video generated with Veo3 (Google DeepMind, 2025).

(ii) localize or summarize long-horizon events that unfold over minutes and hours, and (iii) carry out temporal reasoning that demand ordering, counting, or cross-referencing disjoint moments: capabilities fundamentally out of reach for single-frame systems. The latest closed models such as Gemini-2.5 (Comanici et al., 2025) and GPT-4o (Hurst et al., 2024) as well as open models such as Qwen2.5-VL (Bai et al., 2025), Gemma3 (Team et al., 2025), and InternVL3 (Zhu et al., 2025) all showcase these abilities, and recent benchmarks (Nagrani et al., 2025) confirm that many reasoning tasks truly require video context rather than per-frame cues.

Extending from images to video, however, *amplifies* several long-standing weaknesses. Even for Vision Language Models (VLMs) that process a single image, the image tokens constitute the vast majority of the context length (Yin et al., 2025; Bi et al., 2025). Every additional frame inflates the sequence length, so that the token counts and compute budgets explode. Compute-optimal analyses show that merely feeding more frames is rarely the most efficient use of inference FLOPs (Wang et al., 2025a). Even when memory suffices, the accuracy almost always deteriorate with context length, mirroring the "context-rot" (Hong et al., 2025a) observed for LLMs. Fine-grained kinematics—subtle velocity changes, limb articulation, or repetition count—remain difficult (Hong et al., 2025b; Nagrani et al., 2025). Detailed queries often elicit hallucinated or mis-ordered events (Zhang et al., 2024). Moreover, unlike text-based large language models (LLMs), where deeper inference-time reasoning reliably improves accuracy (Jaech et al., 2024; Wu et al., 2024), recent evidence shows that VideoLLM errors stem primarily from impoverished visual inputs (Upadhyay et al., 2025; Li et al., 2025c; Liu et al., 2025a); adding more "thoughts" offers limited gains unless the model's perceptual bandwidth is improved. Hence, simply investing more *sequential* inference-scaling to a VideoLLM offers sharply diminishing returns; meaningful progress instead demands strategies that enlarge the model's perceptual bandwidth.

To overcome the perceptual bottleneck without aggravating the quadratic cost of longer clips, we introduce **Video Parallel Scaling (VPS)**—an inference-time strategy that trades additional parallel computation for richer visual coverage. Concretely, VPS spawns $J$ independent streams, each sampling a different subset of frames, and fuses their predictions through a weighted aggregation. Because streams are processed concurrently, total FLOPs grow linearly with $J$ while the sequence length[2], and hence the memory footprint, remain unchanged. By this construction, we can leverage more and more visual information by increasing the number of parallel streams, which would have been simply dropped otherwise. As illustrated in Fig. 1, each stream provides *complementary* visual evidence, which, when combined together, yields a correct answer. Further, we theoretically show that VPS offers a way to contract the scaling law (Hoffmann et al., 2022) *without* additional training, by showing that the loss only contracts faster when using uncorrelated subsets of frames.

Through extensive experiments covering different model sizes (2B - 32B), VideoLLM architectures (Qwen2.5-VL (Bai et al., 2025), Gemma3 (Team et al., 2025), InternVL3 (Zhu et al., 2025)), and benchmarks (Video-MME (Fu et al., 2025), EventHallusion (Zhang et al., 2024)), we show that VPS consistently outperforms baselines, resulting in additional improvements with more parallel streams. We further show that VPS scales favorably to other inference-time strategies, and is often orthogonal to the other approaches such that it can be used in harmony.

## 2 BACKGROUND

### 2.1 VIDEO LARGE LANGUAGE MODELS

State-of-the-art VideoLLMs pipe a vision backbone—usually a vision transformer (ViT) (Dosovitskiy et al., 2021) variant with positional encoding (Su et al., 2021; Press et al., 2022)—into a frozen or lightly-tuned LLM head. Typical open models include Qwen-VL 2.5 (Bai et al., 2025), Gemma-3 (Team et al., 2025), and InternVL3 (Zhu et al., 2025). Each attaches a lightweight projection layer that maps per-frame patch embeddings to language tokens, enabling the downstream transformer blocks to treat visual tokens and text tokens uniformly. While open-source VideoLLMs have made rapid progress, comprehensive evaluations show they still hallucinate events, mis-count repetitive actions, and crumble when genuine temporal reasoning is required, especially on long or high-frame-rate clips (Zhang et al., 2024; Hong et al., 2025b; Nagrani et al., 2025; Fu et al., 2025; Li et al., 2025a).

---

[2]This computations is also embarassingly parallelizable.

**Challenges of long context** Significant progress has been made to enable the use of long context in LLMs (Xiong et al., 2024; Chen et al., 2024; Team et al., 2024) to a point where 128k context window is now standard in many proprietary and open checkpoints (Liu et al., 2025b). Nevertheless, feasibility does not imply proficiency. A wide collection of studies points out that the decrease in LLM performance is inevitable with longer context (Li et al., 2025b; Hsieh et al., 2024; Bai et al., 2024; An et al., 2025; Shi et al., 2025; Hong et al., 2025a). Such a finding naturally translates to VideoLLMs, where increasing the number of frames for input leads to a degradation in the performance above a certain threshold (Gao et al., 2025; Wang et al., 2025a), especially for VideoLLMs of modest size. Because every additional frame multiplies the token sequence length, most systems either fix a static budget (e.g. $\leq 32$ frames), or down-sample videos to $\leq 1$ frames per second (fps). On the other hand, we would want to show the model as many video frames as possible, especially when the goal is to comprehend the intricate temporal details. A natural question arises: is this possible without increasing the context length? VPS attempts to give an answer to this question with a positive, namely by constructing parallel streams with different video frames shown for each stream.

## 2.2 INFERENCE-TIME ENHANCEMENT STRATEGIES

We categorize the training-free inference-time enhancement strategies into two: 1) non-scaling approaches that have static compute and gain, 2) scaling approaches that enjoy improved performance when more compute is used.

**Non-scaling approaches** Several frame selection methods have been devised in order to improve upon the usual uniform sampling strategy, to manage the number of frames used for VideoLLMs within a budget. AKS (Tang et al., 2025) and BOLT (Liu et al., 2025c) use query-frame similarity as a measure to select more relevant frames. VideoTree (Wang et al., 2025b) employs a hierarchical structure to select frames for long videos. Modifications to attention have also been devised. SlowFast-LLaVA (Xu et al., 2024) uses two different video frames that are sampled with different fps and combines them with concatenation. Other methods such as modifying the attention, or leveraging external vision models for better interpretation, have also been proposed. SlowFocus (Nie et al., 2024) introduces multiple frequency mixing attention and mixed frequency sampling from relevant segment grounding. MVU (Ranasinghe et al., 2025) proposes to use off-the-shelf vision models to decipher the video into language, then uses the summarized information as the input context. Token merging strategies (Fu et al., 2024; Hyun et al., 2025) attempt to preserve the original accuracy while using a fraction of the tokens.

Methods that operate on the logit-probability level are also popular. Contrastive decoding (CD) (Li et al., 2023; Leng et al., 2024) induces a vector nudge in the logit space by contrasting positive and negative pairs. Temporal contrastive decoding (TCD) (Zhang et al., 2024) extends CD to video sequence. RITUAL (Woo et al., 2024) constructs an augmented view of an image, and sums the logit values before decoding.

**Scaling approaches** Best-of-N (BoN) sampling (Stiennon et al., 2020) selects the answer with the highest reward after running $N$ parallel decoding streams. Speculative rejection (Sun et al., 2024) improves naive BoN by incorporating rejection sampling with reward models during the sampling process. Jinnai et al. (2024) proposes Regularized BoN, and Ichihara et al. (2025) extends this method into a stochastic version, showing theoretical guarantees. While effective, one can only use the variants when having access to a reward function. Self-consistency (Wang et al., 2023) selects the mode of the parallel streams, and hence is free from the dependence on external reward models. On the other hand, Chain-of-Thought (CoT) (Kojima et al., 2022) is a sequential scaling approach that forces the model to reason before answering directly. Tree-of-Thoughts (ToT) (Yao et al., 2023) generalizes CoT through a search process. The advantage of the latter approaches is that they can be used without access to external reward functions. VPS is a method that belongs to the *scaling* category that is free from external dependencies, which scales favorably compared to other parallel scaling methods such as Self-consistency.

While not training-free, ParScale (Chen et al., 2025) is highly relevant to our work, where the authors propose a way to perform prefix tuning for $J$ different streams, so that when decoding the next token, each stream offers a different *view* of the same text. We note that ParScale requires optimizing the

prefix during training. In contrast, VPS is completely training-free, as it is easy to construct different views simply by selecting different frames from the video.

## 3 METHOD

### 3.1 VIDEO PARALLEL SCALING

VideoLLMs take in as input a subsampled version of $T$-frame input video $I_{1:T}$. Define a frame selector function $\mathcal{S}$, which selects $K$ out of $T$ frames.

$$\mathcal{S}_K : \{1, \ldots, T\} \mapsto \{0, 1\}, \quad \mathcal{S}_K(t) = \begin{cases} 1, & \text{if frame } t \text{ is kept,} \\ 0, & \text{otherwise.} \end{cases} \tag{1}$$

The kept indices are $K := \{t | \mathcal{S}_K(t) = 1\} = \{t_1 < \cdots < t_{|K|}\}$. For every frame $I_{t_k}$, embeddings are extracted from the vision encoder $f^\phi$ to get $M$ soft tokens of dimension $d$

$$V = [V_{t_1}, \cdots, V_{t_{|K|}}], \quad V_{t_k} = f^\phi(I_{t_k}) = (v_{t_k}^1, \cdots, v_{t_k}^M) \in \mathbb{R}^{M \times d}. \tag{2}$$

Denote $\Sigma$ as the vocabulary. The probability of the answer $y_{1:N} \in \Sigma^N$ given the prompt $x_{1:L} \in \Sigma^L$ and the vision context $V$ is modeled as

$$p^\theta(y_{1:N} | x_{1:L}, V) = \prod_{n=1}^N p^\theta(y_n | y_{<n}, x_{1:L}, V), \tag{3}$$

where the probability values are obtained through the softmax of logits, i.e.

$$p^\theta(y_t | y_{<t}, x_{1:L}, V) = \text{softmax}(\boldsymbol{z}_t), \quad \boldsymbol{z}_t = g^\theta(y_{<t}, x_{1:L}, V) \in \mathbb{R}^{|\Sigma|}. \tag{4}$$

Assume that we have a set of $J$ frame selection functions $\{\mathcal{S}_{K_1}, \cdots, \mathcal{S}_{K_J}\}$, which would lead to a different set of soft tokens $V_j$ and estimated logit-probability values. Then, our goal is to aggregate the predictions from each stream so that we can incorporate the information from different frames in a collaborative fashion.

$$\bar{p}^\theta(y_t | y_{<t}, x_{1:L}, V) = \sum_{j=1}^J w_j p_j^\theta(y_t | y_{<t}, x_{1:L}, V_j), \quad \boldsymbol{w} = [w_1, \cdots, w_J] \in \Delta^J, \tag{5}$$

where $\Delta^J$ denotes the $J$-simplex[3]. Once a token is sampled, i.e. $y_t \sim \bar{p}_\theta(y_t | \cdot)$, we concatenate the same sampled token to each stream, then iterate, i.e.

$$y_{t+1} \sim \bar{p}^\theta(y_{t+1} | \cdot), \quad \bar{p}^\theta(y_{t+1} | y_{\leq t}, x_{1:L}, V) = \sum_{j=1}^J w_j p_j^\theta(y_{t+1} | y_{\leq t}, x_{1:L}, V_j), \tag{6}$$

until the end of sequence. See Fig. 1 for an illustration. While it is free to choose any selector functions $\mathcal{S}_j$ and weighting functions $w_j$ for VPS, a canonical example would be using uniform sampling for all the streams but with varying offsets, with equal weighting $w_j = 1/J, \forall j$. For instance, for a $T = 64$ frame video with $|K| = 4$ frames and $J = 4$ streams, the following frames would be selected: $K_1 = \{0, 16, 32, 48\}, K_2 = \{4, 20, 36, 52\}, K_3 = \{8, 24, 40, 56\}, K_4 = \{12, 28, 44, 60\}$. Note that dropping $K_{2:4}$ degrades the process to baseline VideoLLM sampling. Throughout the remainder of the manuscript, we use this canonical sampling scheme if not specified otherwise.

### 3.2 THEORETICAL ANALYSIS

Here, we analyze how VPS effectively scales under the lens of Chinchilla scaling law, following the presentation in Hoffmann et al. (2022); Chen et al. (2025). Through the analysis, we reveal how VPS is able to improve the performance of VideoLLMs, as well as deduce desirable strategies for frame sampling in each stream. Throughout the section, with a slight abuse of notation, we denote $x$ as the context, $y$ as the next token label, and assume $w_j = 1/J$ for simplicity. Under this

---

[3]Alternatively, one can also aggregate the logit values. We find that both approaches lead to similar results. See App. B for further discussion

notation, the true next token distribution reads $p(y|x)$. We further let $x_j$ denote the context induced by selecting a subset of frames shown to stream $j$ selected by $\mathcal{S}_{K_j}$. Further, denote $p_j(y|x_j)$ the prediction of stream $j$. The details of the treatment along with the proofs and derivations are deferred to Appendix A.

We first review the simplified version of Chinchilla scaling law (Hoffmann et al., 2022)[4] of LLMs. Concretely, the cross-entropy (CE) loss $L$ of the model with $N$ parameters follows

$$L(N) = E + \frac{A}{N^\alpha}, \tag{7}$$

where $E$ is the irreducible entropy of the natural text, and $A, \alpha$ are constants.

For VideoLLMs, a single stream receives a subset of frames $x_j = \mathcal{S}_{K_j}(x)$ so that every $x_j$ is incomplete in information. Concretely, let

$$p_j(y|x_j) = p(y|x)(1 + \Delta_j), \quad \Delta_j := \frac{p_j - p}{p}, \tag{8}$$

where $\Delta_j$ total relative error that measures the deviation of the stream's imperfect answer $p_j$ from the true answer $p$. The relative error can be split into two parts

$$\Delta_j = \underbrace{\mathbb{E}_{x,y}[\Delta_j]}_{\text{Bias}} + \underbrace{\varepsilon_j}_{\text{Variance}}. \tag{9}$$

The bias is the systematic or predictable part of the error, as stream $j$ always misses a specific set of frames, and the remaining is the variance, with $\mathbb{E}_{x,y}[\varepsilon_j] = 0$ by construction. Further define

$$B_j = -\mathbb{E}_{x,y}[\Delta_j], \quad \varepsilon_j = \Delta_j + B_j. \tag{10}$$

Then, we have the following result

**Proposition 1** (informal). *A VideoLLM that is shown some subset of frames $x_j$ follow*

$$L_j^{\text{VideoLLM}}(N) \approx E + \frac{A}{N^\alpha} + B_j. \tag{11}$$

*Further, the expected CE loss of VPS with $J$ streams follow*

$$L^{\text{VPS}}(N, J) \approx E + \frac{A}{(NJ^{1/\alpha})^\alpha}[1 + (J-1)\rho] + \bar{B}(J), \tag{12}$$

*where $\bar{B}(J) = \frac{1}{J}\sum_{j=1}^J B^{(j)}$, and $\rho$ is the correlation coefficient between $\varepsilon_i, \varepsilon_j, i \neq j$.*

Notice that $L^{\text{VPS}}$ is similar to what was shown in Chen et al. (2025), but with an additional offset $\bar{B}(J)$, which stems from the fact that the VideoLLMs see partial video frames. Two conclusions can be derived.

First, we wish to keep $\rho \in [0, 1]$ as small as possible for the loss to be fast-decaying with $J$. For this, it is crucial that we select *distinct* frames for each stream to maximize diversity. Note that this is different from the data augmentation strategy in Woo et al. (2024), which would yield high $\rho$ as both streams would see the same frames, and thus, small gains even when using parallel streams. In Sec. 4, we empirically show that this is indeed the case.

Second, increasing $J$ does not increase bias unless you pick a subset $x_j$ that yields high KL divergence $\text{KL}(p\|p_j)$. To control this bias term, uniform strides of video frames per stream should be preferred, akin to how VideoLLMs are trained. Notably, both conditions are satisfied when we use our canonical sampling scheme with uniform sampling per stream and constant phase offsets between the streams.

## 4  EXPERIMENTS

**Experimental settings**  We test our method on 3 different model classes, which are considered the state-of-the-art open-source VideoLLMs. For Qwen2.5-VL (Bai et al., 2025), we take 3B,

---

[4]The treatment follows Chen et al. (2025), ignoring the training tokens spent.

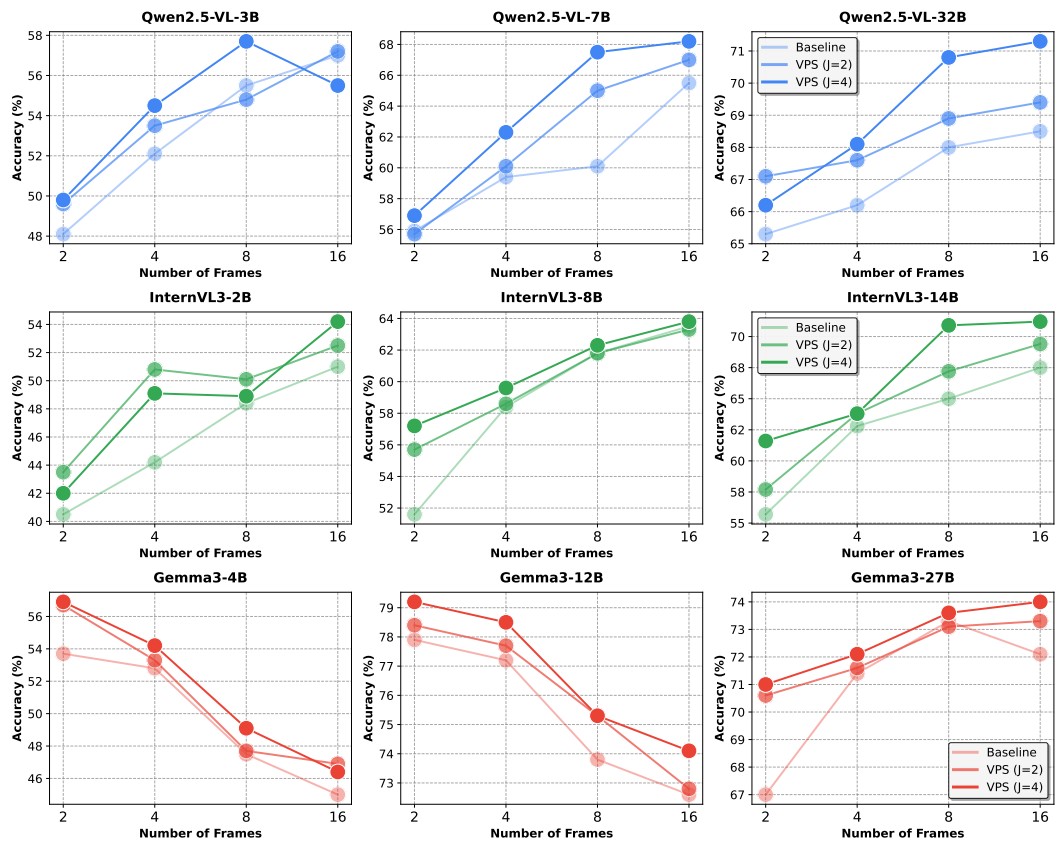

Figure 2: **VPS consistently improves performance across all dimensions**. Across 3 different model classes (Qwen-2.5-VL, InternVL3, Gemma3), 3 different size (2B - 32B), and number of frames used in context, VPS offers improved results with clearer trends with larger models. y-axis denotes the accuracy in the EventHallusion binary QA.

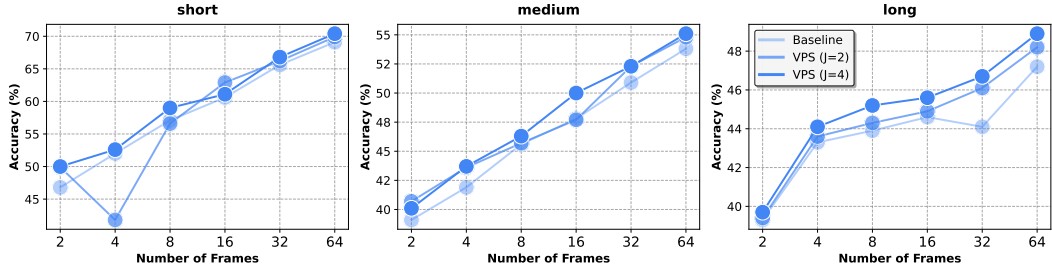

Figure 3: **VPS scales better for longer videos**. Comparing the results of Qwen2.5-VL-7B on Video-MME for each category, we see a clearer trend in the long video category (15 - 30 min.).

7B, and 32B models. For Gemma3 (Team et al., 2025), 4B, 12B, and 27B models are used. For InternVL3 (Zhu et al., 2025), 2B, 8B, 14B models are used. These models are evaluated across 2 different benchmarks: Video-MME (Fu et al., 2025) and EventHallusion (Zhang et al., 2024). Video-MME is a general video understanding benchmark, which consists of 900 videos and 2,700 questions, categorizing the video into short, medium, and long. The length of the video ranges from a few minutes to hours. We consider the case without subtitles. EventHallusion is a benchmark focused on evaluating the hallucination of VideoLLMs, with three categories: entire, mix, and misleading. The benchmark consists of 400 videos and 711 questions, including binary QA and open-ended QA. The duration of the videos range from a few seconds to 30 seconds. Baseline refers to the standard next token sampling with subsampled video frames with a single stream.

Table 1: **Results of VPS on different categories of each benchmark.** VPS improves the performance consistency across categories, regardless of the model. 32 frames are used for Video-MME (Avg. length: ∼ 17 min.) and 8 frames are used for EventHallusion (Avg. length ∼ 20 sec.).

| Model | Method | Video-MME | | | | EventHallusion | | | |
|---|---|---|---|---|---|---|---|---|---|
| | | Short | Medium | Long | Overall | Entire | Misleading | Mix | Overall |
| Qwen2.5-VL-7B (Bai et al., 2025) | Baseline | 0.656 | 0.509 | 0.441 | 0.535 | 0.579 | 0.487 | 0.843 | 0.601 |
| | VPS ($J = 2$) | 0.662 | **0.523** | 0.461 | 0.549 | 0.605 | 0.560 | 0.872 | 0.650 |
| | VPS ($J = 4$) | **0.668** | 0.523 | **0.467** | **0.553** | **0.658** | **0.560** | **0.912** | **0.675** |
| InternVL3-8B (Zhu et al., 2025) | Baseline | 0.719 | 0.570 | 0.506 | 0.598 | **0.421** | 0.616 | 0.843 | 0.618 |
| | VPS ($J = 2$) | **0.742** | 0.570 | 0.506 | 0.598 | 0.404 | 0.622 | **0.852** | 0.619 |
| | VPS ($J = 4$) | 0.728 | **0.603** | **0.533** | **0.622** | 0.412 | **0.632** | 0.843 | **0.623** |
| Gemma3-12B (Team et al., 2025) | Baseline | 0.509 | 0.466 | 0.438 | 0.471 | 0.605 | 0.803 | 0.824 | 0.753 |
| | VPS ($J = 2$) | 0.513 | **0.470** | **0.454** | 0.479 | 0.612 | 0.812 | 0.829 | 0.761 |
| | VPS ($J = 4$) | **0.523** | 0.466 | 0.452 | **0.480** | **0.623** | **0.824** | **0.833** | **0.770** |

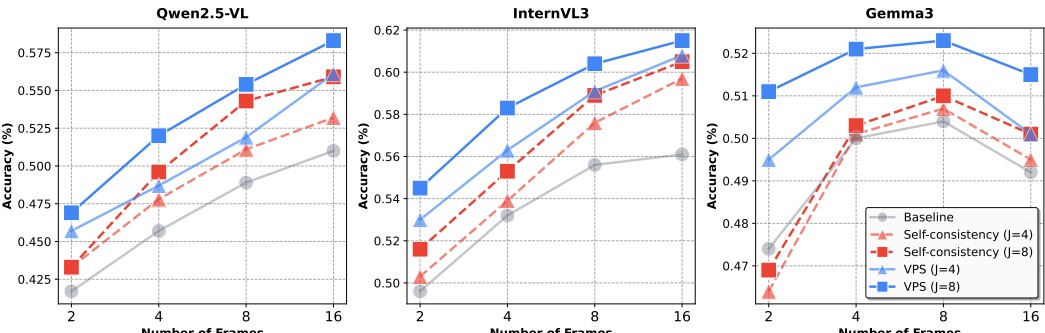

Figure 4: **VPS scales favorably compared to Self-consistency.** On Video-MME, VPS outperforms Self-consistency under the same budget by being able to incorporate information from different frames, rather than relying on the same information for all the streams.

**VPS consistently improves performance across all dimensions** We test the performance of VPS by varying the number of frames used in context (2 - 16 frames for EventHallusion, 2 - 32 frames for Video-MME). In Fig. 2 we see that VPS improves the performance of VideoLLMs as we increase the number of streams ($J$) used, regardless of the number of frames, the model class, and the scale. Due to the quadratic complexity of the attention operation, and the high token count of video frames, moderate-sized VideoLLMs easily face out-of-memory (OOM) issues. For instance, Qwen2.5-VL-32B model with 32 frames results in OOM on a A6000 GPU. In contrast, VPS offers a way of scaling compute to yield better performance with *constant* memory, achieving improvements by increasing $J$. When exceeding a certain budget, the performance of VideoLLMs tend to either plateau or decrease as one incorporates more frames into the context, as can also be observed in the plot in Fig. 2. VPS provides a viable alternative especially in this regime, by being able to scale the results with more compute in the parallel direction[5]. In Appendix Fig. 5 we see a similar trend for Video-MME. Experimental details are provided in Appendix C.1.

In Tab. 1 we provide a more detailed view into how VPS performs across different categories in each benchmark, where we see consistent improvements regardless of the category. Notice that we fixed the used frame count to 32 for Video-MME and 8 for EventHallusion, considering the average length of the videos. We further elucidate the results according to the different length and frame counts used in Fig. 3, where we show that the scaling behavior is increasingly well observed as the length of the video increases. Interestingly, the improvement from VPS becomes more pronounced as model capacity grows and as video duration increases (Fig. 2, 3), a direction where the community is already pushing.

**VPS scales favorably compared to Self-consistency** Self-consistency (Wang et al., 2023) is one of the few methods that allows parallel scaling without the reliance on external modules, e.g. reward functions. A natural questiona arises: Does VPS scale better than Self-consistency? In Fig. 4, we

---
[5]This is especially evident with Gemma3, as can be seen in the bottom row of Fig. 2.

Table 2: **VPS outperforms baseline on free-form description task.** LLM-as-a-judge score in Likert scale, sentence similarity, and ROUGE-L scores are reported on the EventHallusion description task.

| Model | Method | Nframe = 2 | | | 4 | | | 8 | | | 16 | | |
|---|---|---|---|---|---|---|---|---|---|---|---|---|---|
| | | LLM | STS | ROUGE | LLM | STS | ROUGE | LLM | STS | ROUGE | LLM | STS | ROUGE |
| Qwen2.5-VL-7B | Baseline | 2.06 | 44.3 | 19.8 | 2.41 | 48.0 | 19.2 | **2.50** | 49.2 | 19.7 | 2.05 | 43.5 | 19.2 |
| (Bai et al., 2025) | VPS ($J=4$) | **2.25** | **46.1** | 19.8 | **2.47** | **48.7** | **19.5** | 2.43 | **49.8** | **19.9** | **2.52** | **50.8** | **19.8** |
| InternVL3-8B | Baseline | 2.09 | 41.9 | 9.24 | 2.18 | 42.1 | **9.14** | 2.30 | 43.0 | **9.26** | 2.24 | 44.5 | 9.20 |
| (Zhu et al., 2025) | VPS ($J=4$) | **2.14** | **43.7** | **9.89** | **2.23** | **43.0** | 9.10 | **2.39** | **45.0** | 9.24 | **2.33** | **44.8** | **9.29** |
| Gemma3-12B | Baseline | 1.94 | 45.8 | 18.0 | **2.26** | 48.2 | **17.8** | **2.51** | 48.7 | 17.3 | 2.33 | 48.4 | 17.9 |
| (Team et al., 2025) | VPS ($J=4$) | **2.05** | **47.5** | **18.7** | 2.21 | **48.4** | 17.5 | 2.49 | **48.9** | **18.8** | **2.40** | **48.5** | **18.5** |

**EventHallusion**

| Video |  |
|---|---|
| Baseline | The video captures a motorcycle accident where a car and a motorcycle collide, resulting in the motorcycle being flipped over, while pedestrians nearby assess the situation on a rural road. |
| VPS ($J=4$) | The video captures a collision between a car and a three-wheeled vehicle, resulting in significant damage to both vehicles, as seen from the perspective of a car on the road behind them. |

**Video-MME**

| Video |  |
|---|---|
| Caption | Who ultimately won the high jump competition in the video? 
 Baseline:     A. Athlete wearing a white top and black trousers. 
                B. Athlete wearing a white top and white shorts. 
 VPS ($J=4$):   C. Athlete wearing a yellow top and green shorts. 
                D. Athlete wearing a yellow top and black trousers. |

Table 3: **Qualitative analysis of VPS.** VPS offers advantages in both free-form answering tasks, and mulitple-choice QA.

answer with a positive by showing that this is indeed the case, with the details on the experiments provided in Appendix C.2. Notice that on a multiple-choice QA, where the model is forced to answer with a single token, the two strategies are equivalent with large enough sample size: 1) aggregating the probabilities from different streams, 2) sampling from each stream, and determining the answer through majority voting. In this regard, the only difference between Self-consistency and VPS in this constrained situation is whether the different streams $j$ sees different frames of video. Fig. 4 emphasizes the importance of using different frames for each stream by showing superior scaling performance for VPS.

**Free form answering**     Another advantage of VPS is that it can be used for free form answering such as captioning, whereas methods such as Self-consistency cannot. We test the capability of VPS on the description task of EventHallusion (Zhang et al., 2024), and evaluate the performance using LLM-as-a-judge (Zheng et al., 2023) following Zhang et al. (2024), sentence similarity (STS), and ROUGE-L scores. We use `Gemini-2.5-flash` for computing the LLM-as-a-judge score, and use the SentenceTransformer library from huggingface (Model: `all-MiniLM-L6-v2`) to compute STS. Experimental details are provided in Appendix C.3. We see in Tab.2 that VPS consistently outperforms the vanilla baseline in most metrics, with a qualitative example in Tab. 3, illustrating how VPS successfully mitigates the hallucination caused in the baseline decoding strategy.

**Incorporation of other decoding strategies**     There exists many different strategies that aim to enhance the decoding results zero-shot, at the expense of additional parallel computational overhead, similar to VPS. For instance, TCD (Zhang et al., 2024) extends contrastive decoding to videos, by constructing a negative stream where half of the frames are zeroed-out in an interleaving fashion. RITUAL (Woo et al., 2024) constructs an augmentation of the visual frame for collaborative decoding. Implementation details can be found in App. C.4. Is VPS better than these methods? More importantly,

Table 4: **VPS offers complementary improvements.** VPS can be used together with existing inference-time decoding strategies. The same number of frames are used as in Tab. 1, with Qwen2.5-VL-7B as the base model.

| Method | Video-MME | | | | EventHallusion | | | |
|---|---|---|---|---|---|---|---|---|
| | Short | Medium | Long | Overall | Entire | Misleading | Mix | Overall |
| Baseline | 0.656 | 0.509 | 0.441 | 0.535 | 0.518 | 0.508 | 0.725 | 0.565 |
| + VPS ($J = 4$) | **0.668** | **0.523** | **0.467** | **0.553** | **0.605** | **0.560** | **0.873** | **0.650** |
| TCD | 0.651 | 0.510 | 0.453 | 0.538 | 0.588 | 0.497 | 0.863 | 0.614 |
| + VPS ($J = 4$) | **0.668** | **0.547** | **0.479** | **0.564** | **0.605** | **0.576** | **0.882** | **0.662** |
| RITUAL | 0.652 | 0.512 | 0.455 | 0.540 | 0.520 | 0.510 | 0.762 | 0.572 |
| + VPS ($J = 4$) | **0.665** | **0.535** | **0.472** | **0.559** | **0.601** | **0.579** | **0.885** | **0.655** |

Table 5: **Frame sampling strategy** for VPS on EventHallusion with Qwen2.5-VL-7B depending on the number of frames. Uniform: sampling strategy used in the rest of the experiments.

| Method | 2 | 4 | 8 | 16 |
|---|---|---|---|---|
| Baseline | 0.559 | 0.594 | 0.660 | 0.655 |
| Dense ($J = 4$) | 0.546 | 0.587 | 0.665 | 0.658 |
| BOLT ($J = 1$) | 0.579 | 0.612 | 0.612 | 0.645 |
| BOLT ($J = 4$) | **0.581** | **0.630** | 0.638 | 0.643 |
| Uniform ($J = 4$) | 0.569 | 0.623 | **0.675** | **0.682** |

can we use VPS in unison with these other approaches? In Tab. 4, we answer both of these questions with an affirmative. Here, we show that using other decoding strategies such as TCD and RITUAL further improves VPS, hinting that the advantage gained through VPS is largely orthogonal to the previous approaches.

**Choices on frame sampling** The design space of VPS includes the frame selector function $\mathcal{S}$ in (1). In Sec. 3.2, we argued that the canonical uniform sampling strategy is already a sufficiently good choice. Is this really the case empirically, and is there a better strategy? In Tab. 5, we provide answers to these questions. See App. C.5 for experimental details. First, we observe that dense sampling is largely inferior due to the excessive bias $B_j$ this yields. Second, when the number of frames used per stream is small compared to the video length, incorporating other frame sampling strategies such as BOLT (Liu et al., 2025c) yields further improvement, as this strategy lets the VideoLLM attend to more relevant information, decreasing the bias $B_j$. Finally, when the number of frames used per stream is sufficient, the canonical sampling strategy is the winner.

## 5 CONCLUSION

In this work, we introduced Video Parallel Scaling (VPS), a training-free inference-time method designed to overcome the perceptual limitations of VideoLLMs. The core challenge for these models is that increasing the number of input frames for finer temporal understanding leads to prohibitive computational costs and performance degradation. VPS addresses this by processing multiple, disjoint subsets of video frames in parallel streams. By aggregating the output probabilities from these complementary views, VPS effectively expands the model's perceptual bandwidth without increasing the context length or memory footprint of any single stream. Our theoretical analysis demonstrates that VPS contracts the Chinchilla scaling law by leveraging uncorrelated visual information from parallel streams. Extensive experiments across a diverse range of models (from 2B to 32B parameters), architectures, and benchmarks consistently validate our approach. We show that VPS provides significant performance gains, scales more favorably than alternatives like Self-consistency, and is orthogonal to other decoding strategies, allowing it to be used in concert for even greater improvements. These results establish VPS as a robust and memory-efficient framework for enhancing the temporal reasoning capabilities of modern VideoLLMs.

**Limitations and future directions** While VPS demonstrates consistent improvements, its current implementation presents opportunities for future refinement. Presently, VPS applies a uniform weighting to the output of each parallel stream. A promising avenue for future work would be to develop a dynamic weighting scheme. For instance, streams could be weighted based on an information-theoretic measure like the entropy of their output distributions, potentially prioritizing more "confident" or informative views (Farquhar et al., 2024). Furthermore, the current aggregation method is a simple summation of probabilities. Since each stream observes a different phase offset of the same underlying video, a more sophisticated fusion mechanism could yield substantial benefits. Exploring aggregation schemes akin to a multi-agent debate (Du et al., 2023), where streams can interact or influence each other's outputs before a final decision, could lead to a more nuanced and accurate final prediction.

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

## A    PROOFS

In Chen et al. (2025), the inputs to the parallel streams are learnable transformation of the same input $x$, so that one can assume that each parallel stream follows (7) in an unbiased way, leading to a simplification in the analysis of the parallel scaling law. We start by reviewing the result from Chen et al. (2025).

**Lemma 1** (Chen et al. (2025)). *The loss of ParScale with $J$ streams, each indicated with $p_j$, follow*

$$L^{\text{ParScale}}(N, J) = E + \frac{A}{(NJ^{1/\alpha})^\alpha} D + \mathcal{O}(\Delta_j^3), \quad D := (J-1)\rho + 1, \tag{13}$$

*where $\Delta_j := \frac{p_j - p}{p}$. $D$ depends on the correlation $\rho$ of the residuals between the streams. The loss decays fastest when $\rho = 0$, and degrades to the original Chinchilla law when $\rho = 1$.*

*Proof.* For each stream, following (7), the CE loss reads

$$L_i = \mathbb{E}_{x,y}[-\log p_j(y|x)] \tag{14}$$

$$= \mathbb{E}_{x,y}[-\log\{p \cdot (1 + \Delta_j)\}] \tag{15}$$

$$= \underbrace{\mathbb{E}_{x,y}[-\log p]}_{=:E} + \underbrace{\mathbb{E}_{x,y}[-\log(1 + \Delta_j)]}_{A/N^\alpha}, \tag{16}$$

which can be deduced by comparing the loss with (7). Using the Taylor expansion $\log(1 + x) = x - \frac{x^2}{2} + \mathcal{O}(x^3)$, we can expand the second term of the rhs

$$\mathbb{E}_{x,y}\left[\Delta_j - \frac{\Delta_j^2}{2} + \mathcal{O}(\Delta_j^3)\right] = \mathbb{E}_{x,y}\left[\frac{p_j - p}{p}\right] + \mathbb{E}_{x,y}\left[\frac{\Delta_j^2}{2}\right] + \mathcal{O}(\Delta_j^3). \tag{17}$$

Since $\mathbb{E}_{x,y}[p_j] = p$, we have that

$$\mathbb{E}_{x,y}\left[\Delta_j^2\right] = 2A/N^\alpha + \mathcal{O}(\Delta_j^3). \tag{18}$$

Now, we are ready to compute the loss $L$ by averaging the streams. Let $\Delta := \frac{1}{J}\sum_{j=1}^J \Delta_j$. The loss reads

$$L^{\text{ParScale}}(N, J) = E + \mathbb{E}_{x,y}[-\log(1 + \Delta)] \tag{19}$$

$$= E + \mathbb{E}_{x,y}\left[\frac{\Delta^2}{2}\right] + \mathcal{O}(\Delta^3) \tag{20}$$

$$= E + \frac{1}{2J^2}\mathbb{E}_{x,y}\left[\sum_{j=1}^J \Delta_j^2 + 2\sum_{j<k} \Delta_j\Delta_k\right] + \mathcal{O}(\Delta_j^3) \tag{21}$$

$$\overset{(18)}{=} E + \frac{A}{JN^\alpha} + \frac{1}{J^2}\mathbb{E}_{x,y}\left[\sum_{j<k} \Delta_j\Delta_k\right] + \mathcal{O}(\Delta_j^3) \tag{22}$$

$$= E + \frac{A}{JN^\alpha} + \frac{1}{J^2} \cdot J(J-1) \cdot \rho\frac{A}{N^\alpha} + \mathcal{O}(\Delta_j^3) \tag{23}$$

$$= E + \frac{A}{(NJ^{1/\alpha})^\alpha}[(J-1)\rho + 1] + \mathcal{O}(\Delta_j^3) \tag{24}$$

$$\square$$

We are now ready to derive our proposition.

**Proposition 1.** *A VideoLLM that is shown some subset of frames $x_j$ follow*

$$L_j^{\text{VideoLLM}}(N) = E + \frac{A}{N^\alpha} + B_j + \mathcal{O}(B_j^2) + \mathcal{O}(\Delta_j^3). \tag{25}$$

*Further, the expected CE loss of VPS with $J$ streams follow*

$$L^{VPS}(N, J) = E + \frac{A}{(NJ^{1/\alpha})^\alpha}[1 + (J-1)\rho] + \bar{B}(J) + \mathcal{O}(\bar{B}(J)^2) + \mathcal{O}(\Delta_j^3), \tag{26}$$

*where $\bar{B}(J) = \frac{1}{J}\sum_{j=1}^J B_j$, and $\rho$ is the correlation coefficient between $\varepsilon_i, \varepsilon_j, i \neq j$.*

*Proof.* First, recall the definition $B_j := -\mathbb{E}_{x,y}[\Delta_j]$, $\varepsilon_j := \Delta_j + B_j$. The per-stream cross-entropy is

$$L_j^{\text{VideoLLM}}(N) = -\mathbb{E}_{x,y}[\log p_j] = E + \mathbb{E}_{x,y}[-\log(1 + \Delta_j)]. \tag{27}$$

In order for us to use Taylor approximation, first note that

$$\mathbb{E}_{x,y}[\Delta_j] = \mathbb{E}_{x,y}[\varepsilon_j - B_j] = -B_j, \tag{28}$$

as $B_j$ is a constant w.r.t. the expectation in $p(x, y)$. Further, we have that

$$\mathbb{E}_{x,y}[\Delta_j^2] = \mathbb{E}_{x,y}\left[(\varepsilon_j - B_j)^2\right] \tag{29}$$

$$= \mathbb{E}_{x,y}[\varepsilon_j^2] - 2\mathbb{E}_{x,y}[\varepsilon_j B_j] + \mathcal{O}(B_j^2) \tag{30}$$

$$= \mathbb{E}_{x,y}[\varepsilon_j^2] + \mathcal{O}(B_j^2) \quad (\because \mathbb{E}[\varepsilon_j] = 0) \tag{31}$$

Substituting these back into the loss equation leads to

$$L_j^{\text{VideoLLM}}(N) = E + \frac{1}{2}\mathbb{E}[\varepsilon_j^2] + B_j + \mathcal{O}(B_j^2) + \mathcal{O}(\Delta_j^3) \tag{32}$$

$$\overset{(18)}{=} E + \frac{A}{N^\alpha} + B_j + \mathcal{O}(B_j^2) + \mathcal{O}(\Delta_j^3), \tag{33}$$

where we used the result from Lemma 1 to substitute for the variance of the residual. We have completed the first part of the proof.

For VPS, let $\bar{\varepsilon} := \sum_j \varepsilon_j / J$. We have

$$\mathbb{E}_{x,y}[\Delta] = \mathbb{E}_{x,y}[\bar{\varepsilon} - \bar{B}] = -\bar{B} \tag{34}$$

and

$$\mathbb{E}_{x,y}[\Delta^2] = \frac{1}{J^2}\mathbb{E}\left[\left(\sum_{j=1}^{J} \Delta_j^2\right)\right] \tag{35}$$

$$= \frac{1}{J^2}\mathbb{E}\left[\sum_{j=1}^{J} \Delta_j^2 + 2\sum_{j<k} \Delta_j \Delta_k\right] \tag{36}$$

$$= \frac{1}{J^2}\left(\frac{JA}{N^\alpha} + J\mathcal{O}(B_j^2) + J(J-1) \cdot \rho\sqrt{\mathbb{E}[\Delta_j^2]\mathbb{E}[\Delta_k^2]}\right) \tag{37}$$

$$= \frac{A}{(NJ^{1/\alpha})^\alpha}[1 + (J-1)\rho] + \mathcal{O}(\bar{B}(J)^2) \tag{38}$$

Then, the CE loss for VPS reads

$$L^{VPS}(N, J) = E + \mathbb{E}_{x,y}[-\log(1 + \Delta)] \tag{39}$$

$$= E - \underbrace{\mathbb{E}_{x,y}[\Delta]}_{\overset{(34)}{=} -\bar{B}} + \frac{1}{2}\mathbb{E}_{x,y}[\Delta^2] + \mathcal{O}(\Delta^3) \tag{40}$$

$$\overset{(38)}{=} E + \frac{A}{(NJ^{1/\alpha})^\alpha}[1 + (J-1)\rho] + \bar{B}(J) + \mathcal{O}(\bar{B}(J)^2) + \mathcal{O}(\Delta^3) \tag{41}$$

$$\square$$

# B  FURTHER RESULTS

**Probability and logit averaging**  In Tab. 6, we compare the results of logit averaging and probability averaging when implementing VPS. Across different model classes, we find that both approaches lead to similar results. Thus, while we assume probability averaging in the theoretical analysis for simplicity, we resort to logit averaging for implementation.

Table 6: **Results of VPS with logit averaging vs. probability averaging.** Both choices lead to similar results. Small: smallest variant (3B, 2B, 4B) / Base: base variant (7B, 8B, 12B).

| | $J$ | Method | Qwen2.5-VL | | | | InternVL3 | | | | Gemma3 | | | |
|---|---|---|---|---|---|---|---|---|---|---|---|---|---|---|
| | | | 2 | 4 | 8 | 16 | 2 | 4 | 8 | 16 | 2 | 4 | 8 | 16 |
| **Small** | 2 | logit | 0.496 | **0.535** | **0.548** | 0.572 | **0.435** | **0.508** | **0.501** | **0.525** | **0.567** | 0.533 | 0.477 | 0.469 |
| | | prob | **0.498** | 0.523 | 0.540 | **0.577** | 0.433 | 0.489 | 0.491 | 0.523 | 0.550 | 0.521 | **0.491** | **0.472** |
| | 4 | logit | 0.498 | 0.545 | **0.577** | 0.555 | 0.420 | **0.491** | 0.489 | 0.542 | **0.569** | **0.542** | 0.491 | 0.464 |
| | | prob | **0.511** | 0.545 | 0.564 | **0.596** | **0.423** | 0.474 | **0.496** | **0.545** | 0.548 | 0.526 | **0.513** | **0.486** |
| **Base** | 2 | logit | 0.557 | 0.601 | **0.650** | 0.670 | 0.557 | 0.586 | **0.618** | 0.633 | **0.784** | 0.777 | 0.753 | **0.728** |
| | | prob | 0.570 | **0.633** | 0.648 | **0.689** | 0.557 | **0.587** | 0.616 | 0.633 | 0.782 | **0.779** | **0.755** | 0.725 |
| | 4 | logit | 0.569 | 0.623 | **0.675** | **0.682** | 0.572 | **0.596** | 0.623 | 0.638 | 0.792 | 0.785 | 0.753 | **0.741** |
| | | prob | 0.569 | 0.623 | 0.660 | 0.667 | 0.565 | 0.591 | 0.623 | **0.640** | 0.790 | 0.782 | **0.759** | 0.739 |

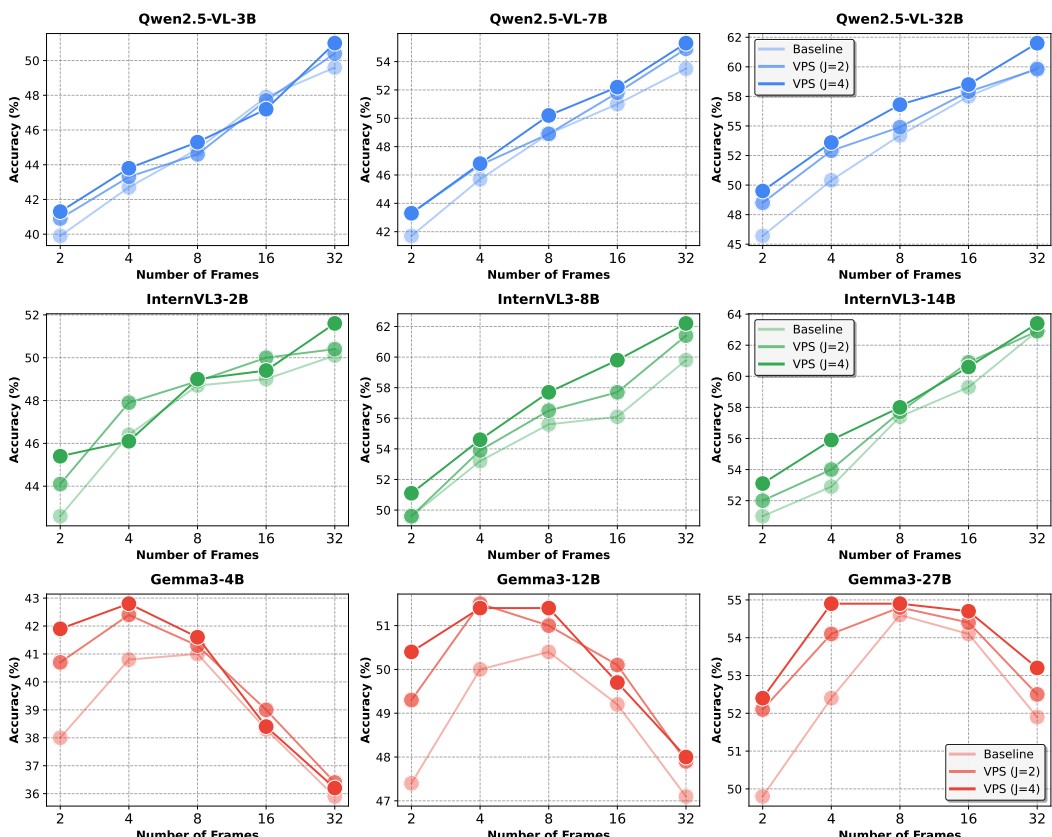

Figure 5: **VPS consistently improves performance across all dimensions**. Across 3 different model classes (Qwen-2.5-VL, InternVL3, Gemma3), 3 different size (2B - 32B), and number of frames used in context. y-axis denotes the accuracy in the Video-MME multiple-choice QA.

**Qualitative results** We present additional examples of the free form description task in Tab. 8.

## C EXPERIMENTAL DETAILS

### C.1 MAIN EXPERIMENT

For Video-MME, following each question, we use the following prompt: "Your response should be a single character: A, B, C, or D. Do not include any other text or explanation.". For EventHallusion, following each question, we use the following prompt: "Please answer yes or no."

## C.2 SELF CONSISTENCY EXPERIMENT

Notice that when decoding a single token, majority voting after decoding, and averaging the probabilities before sampling lead to the same results in the limit of infinite samples. In practice, we notice that due to formatting issues, the performance is slightly better when we use majority voting after the answer is extracted, as such scaffolding generally helps. As the goal in this experiment is to emphasize the importance of using different frames for each stream, we also implement VPS with majority voting, but with seeing different frames for each stream, for fair comparison against Self-consistency. This way, the difference in scaling solely comes from the difference in input frames.

---

**System prompt:**
You are an intelligent chatbot designed for evaluating the correctness of generative outputs for video summaries.
Your task is to compare the predicted answer with the pseudo-reference answer and determine if they match meaningfully.
You should rely more on the video frames than the pseudo-reference caption.
——
INSTRUCTIONS:
- Focus on the meaningful match between the predicted answer and the pseudo-reference answer.
- Consider synonyms or paraphrases as valid matches.
- Evaluate the correctness of the prediction compared to the video frames.
**User prompt:**
Given the reference caption, evaluate the quality of the predicted caption.
Provide your evaluation only as an integer value between 1 and 5, with 5 indicating the highest quality.
Please provide your evaluation in the following format: [evaluation]

---

Table 7: Prompts used for LLM-as-a-judge (Zheng et al., 2023) evaluation system.

## C.3 FREE FORM EXPERIMENT

Since the (pseudo-)ground truth answer in EventHallusion is given in a simple sentence, we use "Summarize the video in one sentence." as the prompt. When using LLM-as-a-judge (Zheng et al., 2023) when evaluating the free form descriptions of the video, we follow Zhang et al. (2024) and use the prompt specified in Tab. 7.

## C.4 INCORPORATING OTHER STRATEGIES

For TCD, we construct a negative stream so that the half the frames are zeroed-out in an interleaved fashion. Let $x'$ be the frame-dropped version of the sub-sampled video. Then, TCD is implemented with

$$\tilde{p}^\theta(y|x) = (1 + \alpha)p^\theta(y|x) - \alpha p^\theta(y|x'), \tag{42}$$

where $\alpha \in [0, 1)$ is a constant. Additionally, we set a hyperparameter $\beta \in [0, 1]$ that keeps only the high probability tokens that exceeds the value $\beta \max_w p^\theta(y|x)$, following (Li et al., 2023). We use the default values $\alpha = 0.5, \beta = 0.1$. When used together with VPS, we use $\bar{p}^\theta$ instead of $p^\theta$. For RITUAL, we consider the following 5 augmentations: horizontal flip, vertical flip, 180 degrees random rotation, color jitter, and gaussian blur. We apply the same augmentation to each frame, and use equal weighting for the original view and the augmented view. When used together with VPS, we construct an augmented view per VPS stream.

## C.5 FRAME SAMPLING STRATEGY

The dense sampling strategy with $J$ streams first makes $J$ chunks from the video sequence. Within the chunk, the frames are uniformly sampled. BOLT (Liu et al., 2025c) computes the CLIP similarity

| Video |  |
|---|---|
| Baseline | A cyclist crashes while riding at night, resulting in her bicycle falling over. |
| VPS ($J = 4$) | A cyclist is struggling to carry her bike down a street. |
| Video |  |
| Baseline | A parrot curiously interacts with a cup of tea and a spoon, attempting to participate in a human's tea-drinking ritual. |
| VPS ($J = 4$) | A parrot curiously tries to stir a cup of tea with a spoon. |

Table 8: **Qualitative analysis of VPS.** VPS effectively captures the overall motion depicted in videos from the EventHallusion dataset.

between the video frames and the query prompt. A normalized score $s_i$ is then sharpened with

$$s_i^r = \left( \frac{s_i - \min(s)}{\max(s) - \min(s)} \right)^\alpha \tag{43}$$

, where $\alpha = 3.0$. We then sample the frames according to this sharpened distribution. When using BOLT together with VPS, it is important that we do not sample the same frames for each stream. Hence, we eliminate the frame indices once the frame is sampled from one of the streams, then sample from the truncated distribution.

# D  LLM USAGE

Large language models were not used for research ideation, methodological design, data analysis, or interpretation. They were used only for minor language polishing. All content was produced and verified by the authors.

