# OpenReview forum: "Video Parallel Scaling: Aggregating Diverse Frame Subsets for VideoLLMs"
_ICLR.cc/2026/Conference — ICLR 2026 Conference Withdrawn Submission_

### Official Review · Reviewer_mnjg · 2025-10-30

**Soundness:** 3
**Presentation:** 2
**Contribution:** 2
**Rating:** 4
**Confidence:** 3

**Summary:**

The paper proposes an inference-time strategy, Video Parallel Scaling (VPS), which expands a model’s perceptual bandwidth without increasing its context window. VPS spawns multiple independent streams, each sampling a different subset of frames to provide complementary visual cues, and then aggregates their predictions afterward to generate the final answer. The approach is validated on two benchmarks using multiple backbones, demonstrating the effectiveness of VPS compared to the baseline. The paper also includes ablation studies on different decoding and frame sampling strategies, further highlighting the effectiveness of the proposed approach.

**Strengths:**

The paper is well organized and easy to read. It presents an interesting inference-time scaling strategy that boosts the performance of long-video understanding while maintaining constant memory usage, which could benefit real world on-device video understanding applications. The paper also includes solid theoretical analysis of parallel stream scaling and provides comprehensive experiments demonstrating the effectiveness of the proposed approach.

**Weaknesses:**

1. The claim that increasing the number of input frames to capture fine-grained temporal details leads to performance degradation (L031–033, L066–L067) seems somewhat counterintuitive and inconsistent with empirical experience, such as the results shown in Table 9 of [1]. It would be helpful if the authors could provide references or related works supporting this claim. The results of Qwen2.5-VL and InternVL-3 in Figure 4 suggest that performance generally improves as more frames are included. Although Gemma-3 shows a performance drop when increasing the number of frames from 8 to 16, I doubt whether Gemma-3 is an appropriate backend model for long-video understanding tasks, given its much lower performance compared to Qwen2.5-VL and InternVL-3. It might also be useful to experiment with stronger backend models such as GLM-4.1/4.5 [2]
2.	Since each stream is uniformly sampled and equally weighted, I have doubts about the effectiveness of this approach when the streams contain very different content, leading to divergent probabilities and inconsistent results. For example, if some streams are sampled from background while others focus on the foreground, the model may produce inconsistent attention and descriptions when interpreting foreground object actions.
3. It would be beneficial to include evaluations on more challenging benchmarks, such as Long Video Bench, EgoSchema, and MLVU, which involve complex temporal reasoning, to better demonstrate the effectiveness of VPS

[1] Hu, K., Gao, F., Nie, X., Zhou, P., Tran, S., Neiman, T., Wang, L., Shah, M., Hamid, R., Yin, B. and Chilimbi, T., 2025. M-LLM based video frame selection for efficient video understanding. In Proceedings of the Computer Vision and Pattern Recognition Conference (pp. 13702-13712).

[2] V Team, Hong, W. and Yu, W., 2025. Glm-4.5 v and glm-4.1 v-thinking: Towards versatile multimodal reasoning with scalable reinforcement learning. arXiv preprint arXiv:2507.01006.

**Questions:**

1. How does the VPS workflow operate? Does it run sequentially on a single GPU for each stream, or in parallel across multiple GPUs? If it is the former, could the authors provide any insights into the overall latency performance compared to the baseline?
2. The baseline results of Qwen2.5-VL-7B and InternVL3-8B reported in Table 1 are much lower than those obtained under the same 32 input frames setting in other works. Could the authors clarify the root cause? For reference, Qwen2.5-VL achieves 60.8 and InternVL-3 achieves 65.6 in Table 1 of [1], while Qwen2.5-VL reports 61.1 in Table 2 of [2] for the Video-MME w/o sub.
3. How does performance scale as the number of streams $J$ increases?

[1] Zou, Y., Jin, S., Deng, A., Zhao, Y., Wang, J. and Chen, C., 2025. AIR: Enabling Adaptive, Iterative, and Reasoning-based Frame Selection For Video Question Answering. arXiv preprint arXiv:2510.04428.

[2] Sun, G., Singhal, A., Uzkent, B., Shah, M., Chen, C. and Kessler, G., 2025. From Frames to Clips: Efficient Key Clip Selection for Long-Form Video Understanding. arXiv preprint arXiv:2510.02262.

---

### Official Review · Reviewer_CokG · 2025-11-01

**Soundness:** 3
**Presentation:** 3
**Contribution:** 3
**Rating:** 6
**Confidence:** 2

**Summary:**

This paper proposes a training-free inference framework Video Parallel Scaling (VPS) that can improve long-video understanding for video large language models. VPS uses a simple but effective mechanims to run multiple parallel inference streams, with each seeing a different subset of the video frames, and then aggregate the token-level distributions into the final output. The authors develops the insights with an analysis extending the Chinchilla framework. They show that the combining low-correlation streams can reduce prediction variances and yields faster effective loss contraction.

Compared to prior work, e.g. ParScale, which similarly use multi-stream scaling through leaned prefix tuning, VPS shares similar insights and achieves similar gain but without additional training. Their experiments on Video-MM and EventHallucination across multiple model families demonstrate the consistent improvements in accuracy and reduce the hallucination.

**Strengths:**

The paper is technical sound and simple. The authors demonstrate a simple idea backed up the theory and can be effectively applied to practice easily. The method does not require training, which makes a lot more practical to be used in real world practices.

The authors provide an theoretical analysis to Chinchilla's scaling law with bias-variance decomposition, which is a sound interpretable explanation for why the aggregation of frame subsets can lead improvement and what assumption we need to consider in practice.

The proposed model is model agnostic. Its performance gain evaluated under different family models are consistent and impressive. In addition, the method may also be able to be applied to other inference techniques.

**Weaknesses:**

We may need to scrutinize some assumptions in various real world scenarios, which I am not sure understand in the practical use case well. For videos recorded as different FPS, I am not entirely sure what sampling strategy with combination of strides and frame subsets would be sufficient ensure the framesets are distinct enough while ensuring good phase offsets between the streams. For high frequency video, I would imagine the each frameset would be sufficiently close to each other and leads to high correlations while for low frequency video, choosing large number for frameset would lead to undersampling in each stream and lead to information loss.

In the related work, the authors mention ParScale as the most related work but I don't see a discussion of the evaluation for the two.

**Questions:**

Besides the questions I raised in the weakness part, I am also interested to know when will VPS fail exactly. The authors only vaguely mentions a few improvement of frame sampling in the future work, but I will be more interested to know when will the current strategy fail and why.

From the figure, it seems the increase of J will very likely to better performance but we have not seen when it may saturate.

---

### Official Review · Reviewer_rwEJ · 2025-11-02

**Soundness:** 2
**Presentation:** 3
**Contribution:** 2
**Rating:** 2
**Confidence:** 4

**Summary:**

This paper introduces Video Parallel Scaling (VPS), a training-free inference-time method to enhance the long clip coverage of Video Large Language Models (VideoLLMs). Specifically, instead of feeding longer sequences or more frames into the model, VPS feeds multiple disjoint subsets of video frames in parallel inference streams, which avoids quadratic attention costs. The predictions from these streams are then aggregated in token probabilities to predict an ensemble output. Experiments across several VideoLLMs show improvements on multiple benchmarks over baselines.

**Strengths:**

1. Clear motivation: The paper targets an important problem in video understanding: how to efficiently handle long video coverage without quadratic cost. And it proposes a simple and practical inference-time remedy.
3. Training-free design: VPS can be readily applied to existing VideoLLMs without additional training, which increases its usability.
3. The proposed method is evaluated across a range of model sizes (2B–32B) and benchmarks, and the results are good across architectures and scales.

**Weaknesses:**

1. Lack of sufficient novelty: The main method of this paper is running multiple inference streams on different frame subsets and ensembling their probabilities. The idea is quite straightforward and very similar to existing ensemble methods like self-consistency, best-of-N. The difference is just assigning different sampled frames to each stream, which is a small heuristic variation rather than a fundamentally new paradigm.
2. Lack of strong baselines: The experimental section does not include quantitative comparisons against stronger or conceptually similar baselines that also target frame sampling problems, such as SlowFast-LLava and some attention-based methods (though many works are mentioned in related work).
3. The weighting strategy when ensembling different streams is too straightforward - uniform weighting is used. This is quite an important component in the proposed method, but it is not studied.

**Questions:**

Please see the weaknesses.

---

### Official Review · Reviewer_SuLw · 2025-11-02

**Soundness:** 2
**Presentation:** 2
**Contribution:** 2
**Rating:** 2
**Confidence:** 4

**Summary:**

This paper proposes to extract parallel streams (token sequences) of a long video and average the response logits to enhance the video understanding performance. The author provides a theoretical analysis on why this parallel scaling brings improvement. The method is verified across multiple baseline models with various sizes and the video parallel scaling does improve the baseline model.

**Strengths:**

1. The proposed method is easy-to-use. Although not stated by the author, it can be inferred that this method is orthogonal to other long video understanding methods, such as token reduction/merging, context parallelism, and speculative decoding.
2. The evaluation is robust in terms of baseline models. The author comprehensively evaluated Qwen2.5-VL, InternVL3, and Gemma3, each with different sizes.

**Weaknesses:**

1. This paper **lack details**, especially for method and experiments. Here are some examples: 1) How is the parallelism achieved? By using more GPUs or parallel on the batch dimension? 2) What's your video sampling FPS? 3) What's the configuration of VPS for each experiment? How many frames are extracted for each stream? Is it always 4 as stated in line 203? 4) In Table2, what does the "Nframe" refer to? Is it the K?
2. This paper **lacks evaluation on across different benchmarks**. There are many other long video benchmarks that are commonly used for evaluation, such as LongVideo, VideoChat, and NextQA. The main point here is that evaluating more benchmarks is more convincing and can show the robustness of VSP.
3. The paper **lacks comparison**. The only comparison is made against the baseline model. However, given that there have been many long video understanding works so far, such as Video-XL[1], Streaming Long Video Understanding[2], and LongVLM[3], the author is supposed to provide a comprehensive comparison against some SOTA baselines. Or, if VSP is orthogonal to previous methods, the author can also show that previous SOTAs can be easily combined with VSP and achieve a even better result. Without any discussion, it's not convincing that VSP is effective enough.
4. The paper claims to target at long video understanding, but experiments on scaling are not convincing. For example, in Figure 2, 3, and 4, the maximum number of frames is 16/64. Again, since the sampling fps is not specified, I have no clue about the actual video length. However, since long video understanding now usually refers to near hour videos, the scaling in this paper seems a bit trivial.


[1] Video-XL: Extra-Long Vision Language Model for Hour-Scale Video Understanding, in CVPR 2025
[2] Streaming Long Video Understanding with Large Language Models, in NeurIPS 2024
[3] LongVLM: Efficient Long Video Understanding via Large Language Models, in ECCV 2024

**Questions:**

1. Configurations related questions are in W1
2. Since the output logits come from the weighted sum of multiple streams, do multiple streams actually amplify the answer token logits? Specifically, are there any case studies of the logit distribution before and after the VSP?
3. Is there any efficiency improvement such as reduced latency or FLOPs?

---

### Note · Authors · 2025-11-13

I have read and agree with the venue's withdrawal policy on behalf of myself and my co-authors.